# Intermittent Exercise at Lactate Threshold Induces Lower Acute Stress than Its Continuous Counterpart in Middle-to-Older Aged Men

**DOI:** 10.3390/ijerph19127503

**Published:** 2022-06-19

**Authors:** Taiki Yamamoto, Yukiya Tanoue, Yoshinari Uehara, Yasuki Higaki, Hiroaki Tanaka, Kenji Narazaki

**Affiliations:** 1Faculty of Sports and Health Science, Fukuoka University, Fukuoka 814-0180, Japan; tyamamoto@fukuoka-u.ac.jp (T.Y.); ueharay@fukuoka-u.ac.jp (Y.U.); higaki@fukuoka-u.ac.jp (Y.H.); 2The Fukuoka University Institute for Physical Activity, Fukuoka University, Fukuoka 814-0180, Japan; 3Ritsumeikan-Global Innovation Research Organization, Ritsumeikan University, Shiga 525-0058, Japan; y-tanoue@fc.ritsumei.ac.jp; 4Research Organization of Science and Technology Institute of Advanced Research for Sport and Health Science, Ritsumeikan University, Shiga 525-0058, Japan; 5Center for Liberal Arts, Fukuoka Institute of Technology, Fukuoka 811-0295, Japan

**Keywords:** aerobic exercise, exercise prescription, preventive measures, healthy aging

## Abstract

This study aimed to compare the degree of exhaustion and trophic effects between continuous exercise (CE) and intermittent exercise (IE) at lactate threshold (LT) intensity. Seven healthy men (age: 43–69 years) performed the following three experimental tests in a randomized crossover order: (1) control; (2) CE, performed as a 20-min of cycling at LT intensity; and (3) IE, performed as 20 sets of a one-min bout of cycling at LT intensity with a 30-s rest between every two sets. Heart rate (HR), blood lactate concentration (LA), rating of perceived exertion (RPE), catecholamines, cortisol, growth hormone, insulin-like growth factor (IGF)-1, and brain-derived neurotrophic factor (BDNF) were measured. The sampling timing in each test was as follows: 10 min before the onset of exercise, at the 25%, 50%, and 100% time points of exercise, and at 10 min after exercise. IE was found to be accompanied by a lower degree of exhaustion than CE in measures of HR, LA, RPE, catecholamines, and cortisol. In terms of trophic effects, both of IGF-1 and BDNF increased in CE, while a marginal increase of BDNF was observed in IE. The results indicated that IE induces lower stress than CE, but may not be effective for inducing trophic effects.

## 1. Introduction

The World Health Organization (WHO) reported that the population of older adults (≥60) was 1 billion in 2019, and is rapidly increasing [1]. Aging is known to increase the risk of physical, mental, and social disorders. For example, healthy adults lose approximately 8% of their muscle mass every 10 years after the age of 40 years. Accordingly, the prevalence of sarcopenia at the age of 60 years is approximately twice that at the age of 40 years [2].

To prevent and/or delay the risks, many public health guidelines and studies have recommended that older people increase the amount of physical activity they undertake. Several studies have reported that exercise prevents various degenerative diseases, including sarcopenia and dementia [3,4,5]. In addition, guidelines published by the WHO and United States Department of Health and Human Services recommend that older adults should perform at least 150–300 min of moderate-to-vigorous intensity aerobic exercise per week [1,6]. The lactate threshold (LT) has been widely used to prescribe moderate intensity aerobic exercise for preventive and therapeutic interventions against hypertension, diabetes, cardiovascular disease, and sarcopenia [7,8,9,10,11].

The effects of LT intensity exercise on health are often assessed using acute responses of trophic hormones. For instance, growth hormone (GH) and insulin-like growth factor (IGF)-1 can be examined as myotrophic anabolic hormones in such assessments. IGF-1 has myotrophic, anabolic, insulin-sensitizing, and lipid oxidation effects, and is an important metabolic biomarker for health and exercise benefits [11,12]. In one previous study, serum GH and IGF-1 concentrations were increased by LT intensity exercise [13]. In addition, GH has been reported to stimulate IGF-1 production and increases serum IGF-1 concentration [13]. GH has direct or indirect anabolic effects that are mediated by IGF-1 in older adults [12]. In addition, brain-derived neurotrophic factor (BDNF) is a neurotrophic family protein that is required for cell growth, survival, and cell differentiation of neurons [14]. Increasing BDNF is reported to induce a physiological mechanism through which exercise provides psychological benefits [15], and has been observed in aerobic and resistance exercise [16,17].

Given that many older people are considered to suffer from functional restrictions (with or without their awareness) and have limited tolerance for exercise stimulus, exercise warrants safety precautions, requiring careful and appropriate prescriptions. Because body homeostasis is disturbed by exercise and returned to the basal level after stopping it, inserting rests between exercise, or making exercise intermittent, is considered to be important to avoid overreaching. Although this may be a matter of course while exercising, it is often overlooked in aerobic exercise prescriptions. Even in relatively safe LT intensity exercise, lactate accumulation and heart rate (HR) increases when exercising for a long time without rest have been reported [18]. Additionally, the amount and timing of rest periods between exercise have been found to significantly affect metabolic [19], hormonal [20], and other responses [21] in resistance training, while few studies have addressed the effects of rest in aerobic exercise.

Therefore, the current study aimed to compare responses regarding the degree of exhaustion and trophic effects between continuous and intermittent aerobic exercise using cycle ergometry at LT intensity over the same cycling work volume. To examine this issue, we proposed the following two hypotheses: (1) intermittent exercise causes a lower degree of exhaustion compared with continuous exercise; and (2) intermittent exercise induces acute trophic responses, which are lower than those induced by continuous exercise.

## 2. Materials and Methods

### 2.1. Subjects

Seven subjects volunteered to participate in the study. The following were set as inclusion criteria, with all criteria being met: men aged forty years or older and six months or longer of a regular exercise habit. The following exclusion criteria were also set in advance: having past and current histories of cardiovascular, metabolic, endocrine, and psychiatric diseases, and being smokers. All subjects provided informed consent after the purpose, methods, and significance of the study were explained. The study was approved by the Ethics Committee of Fukuoka University (No. 16-9-02).

### 2.2. Baseline Test

Each subject visited our laboratory on four separate days: the first visit for a baseline test, and the second to fourth visits for the three experimental tests. In the baseline test, each subject underwent a multistage cycle ergometer test after a 15-min rest in the sitting position (Ergomedic 874E; Monark Exercise AB, Vansbro, Sweden). The subject ran for four min in each stage, with a 1-min rest between two stages.

A blood sample was obtained from the earlobe during each stage of finished timing to measure the blood lactate concentration (LA, Lactate Pro™2 LT-1730; ARKRAY Inc., Kyoto, Japan). The workload of the cycle ergometer was set at 0.4 kp in the initial stage and was increased by 0.2 kp for each subsequent stage until the LA reached 4 mmol/L or the subject was unable to maintain the workload [22]. After the test, LT was determined by five researchers using the Log-Log transformation method [23].

### 2.3. Experimental Tests

The protocol of the experimental tests is summarized in Figure 1. Briefly, subjects performed the following three experimental tests in a randomized crossover order: (1) control (CON); (2) continuous exercise (CE); and (3) intermittent exercise (IE). The CON comprised a 20-min period of sitting on a standard chair. CE was performed as a 20-min bout of cycling on the cycle ergometer at LT intensity. In contrast, IE was performed as 20 sets of a one-min bout of cycling on the cycle ergometer at LT intensity with a 30-s rest between every two sets. During the rest period in the IE, subjects sat on the cycle ergometer. The regimen for the intermittent exercise was selected based on previous studies in our laboratory [24,25]. During each experiment test, subjects were allowed to drink a moderate amount of water upon request. The experimental tests were performed at least 48 h apart. Subjects were restricted from undertaking strenuous exercise and were not allowed to consume caffeinated drinks and alcohol for 24 h before the tests. Subjects were also directed to come to the laboratory after they fasted for more than 10 h.

### 2.4. Measurements

During each experimental test, sampling timings for measurements were set as follows: 10 min before the onset of exercise (denoted as Rest), at 25%, 50%, and 100% of the cycling interval (denoted as E25%, E50%, and E100%), and at 10 min after cycling (denoted as A10 min) (Figure 1). HR was observed by a HR monitor (HR, RS800CX; PO-LAR., Kempele, Finland) at 20 s, 15 s, 10 s, and 5 s before and just at each sampling timing, and the mean three intermediate measures (after excluding the minimum and maximum measures) was calculated. Borg rating of perceived exertion (RPE) was reported by the subject only during cycling (i.e., at E25%, E50%, and E100%) using the Borg scale (range: 6–20) [26]. For each experimental test, peripheral cannulation was administered at least 15 min before the first blood collection at Rest, in order to exclude the effects of pain on catecholamine concentrations [27]. A blood sample was collected using a syringe at each sampling timing. Immediately after each collection, the blood sample was divided into serum (Plain tube; Terumo Corporation, Tokyo, Japan) and plasma (EDTA-2K tube; Terumo Corporation, Tokyo, Japan) blood-collection tubes. Additionally, the LA was rapidly measured using the remaining blood sample in the syringe. After finishing the experimental test, the serum and plasma tubes were immediately centrifuged at 3000 rpm for 15 min at 4 °C. The supernatants were then transferred to polypropylene tubes and stored at −80 °C until analysis.

### 2.5. Blood Analyses

Blood concentrations of lactate, catecholamines (including adrenaline, noradrenaline, and dopamine), cortisol, GH, IGF-1, and BDNF in the collected blood samples were analyzed. Biochemical analyses for all the indices except lactate were blindly conducted by SRL Inc. (Tokyo, Japan). Specifically, concentrations of catecholamines were measured using high-performance liquid chromatography (CA Test TOSOH; Tosoh Corporation, Tokyo, Japan), those of cortisol, GH, and IGF-1 were measured using electrochemiluminescence immunoassay (cortisol: Elecsys^®^ Cortisol; Roche Diagnostic Systems, Basel, Switzerland, GH: Elecsys^®^ hGH; Roche Diagnostic Systems, Basel, Switzerland, IGF-1: IGF-I IRMA DAIICHI; Fujirebio Inc., Tokyo, Japan), and the concentration of BDNF was measured using an enzyme-linked immunosorbent assay (Quantikine^TM^ ELISA Human Free BDNF Immunoassay; R&D Systems Inc., Minneapolis, MN, USA). As mentioned above, LA was measured at each blood collection using an enzyme electrode method (Lactate Pro™2 LT-1730; ARKRAY Inc., Kyoto, Japan).

### 2.6. Statistical Analyses

Measured values are shown as means ± standard deviation (SD) for all of the indices. Furthermore, to examine intra-subject changes by controlling for inter-subject variability in blood levels [28], catecholamines, cortisol, GH, IGF-1, and BDNF concentrations at E25%, E50%, E100%, and A10 min were normalized relative to those when at Rest. Values of HR, RPE, and blood samples were compared using two-way repeated-measures analysis of variance (ANOVA) for [type] × [time] interactions. The assumption of sphericity was tested via Mauchly’s test of sphericity, and as appropriate, the Greenhouse-Geisser correction was applied. When the ANOVA showed a significant interaction, post-hoc Tukey’s multiple comparison tests [type] and Dunnett’s tests [time] were performed, as appropriate. All statistical analyses were performed using the Graph Pad Prism version 9.3.0 software package (Graph Pad Software., San Diego, CA, USA). The significance level was set at *α* = 0.05.

## 3. Results

### 3.1. Physiological Characteristics

Seven healthy middle-to-older aged men successfully completed the baseline and experimental tests. The physical and physiological characteristics of the subjects are shown in Table 1.

### 3.2. Comparison of a Degree of Exhaustion (with Conventional Indices)

Figure 2 shows changes in conventional indices regarding a degree of exhaustion during the experimental tests. For HR (Figure 2a), there was a significant interaction (*F*_2.72, 16.32_ = 71.95, *p* < 0.01) and significant main effects for [type] (*F*_1.46, 8.79_ = 83.26, *p* < 0.01) and [time] (*F*_1.91, 11.46_ = 95.83, *p* < 0.01). The values of HR in both the CE and IE were significantly higher than those in the CON at all sampling timings except Rest (*p* < 0.05 for all). In addition, the HR values in the IE were significantly lower than those in the CE at all sampling timings during cycling (i.e., E25%, E50%, and E100%) (*p* < 0.05 for all). Compared with Rest, the HR values were significantly higher at the remaining sampling timings in both the CE and IE (*p* < 0.05 for all).

Regarding LA (Figure 2b), there was a significant interaction (*F*_3.31, 19.88_ = 26.07, *p* < 0.01) and significant main effects for [type] (*F*_1.94, 11.67_ = 64.77, *p* < 0.01) and [time] (*F*_2.18, 13.11_ = 41.66, *p* < 0.01). The values of LA in the CE were significantly higher than those in the CON at all sampling timings except Rest (*p* < 0.05 for all). In addition, the LA values in the IE were significantly higher than those in the CON at E25% and E50%, and significantly lower than those in the CE at E50%, E100%, and A10min (*p* < 0.05 for all). Compared with Rest, the values of LA were significantly higher at the remaining sampling times in both the CE and IE (*p* < 0.05 for all).

Regarding RPE (Figure 2c), the data revealed no significant interaction (*F*_2, 12_ = 0.42, *p* = 0.66) but did reveal significant main effects for [type] (*F*_1, 6_ = 60.04, *p* < 0.01) and [time] (*F*_2, 12_ = 25.14, *p* < 0.01).

### 3.3. Comparison of the Degree of Exhaustion (with Additional Indices Derived from Blood Analyses)

Figure 3 shows changes of additional indices derived from blood analyses, which are related to the degree of exhaustion during the experimental tests. For adrenaline (Figure 3a), there was a significant interaction (*F*_2.90, 17.45_ = 6.62, *p* < 0.05) and significant main effects for [type] (*F*_1.87, 10.96_ = 10.01, *p* < 0.05) and [time] (*F*_2.75, 16.50_ = 15.54, *p* < 0.05). The values of adrenaline at E50% and E100% in the CE were significantly higher than those in the CON (*p* < 0.05 for both), and the value at E100% in the IE was significantly higher than that in the CON (*p* < 0.01). Compared with Rest, the values of adrenaline were significantly higher at E50% and E100% in the CE and at E100% in the IE (*p* < 0.05 for all).

Regarding noradrenaline (Figure 3b), there was a significant interaction (*F*_1.79, 10.87_ = 10.38, *p* < 0.01) and significant main effects for [type] (*F*_1.05, 6.34_ = 14.46, *p* < 0.01) and [time] (*F*_1.70, 10.24_ = 13.18, *p* < 0.01). The values of noradrenaline in both the CE and IE were significantly higher than those in the CON at all sampling times during cycling (*p* < 0.05 for all), and the values at E50% and E100% in the IE were significantly lower than those in the CE (*p* < 0.01 for both). Compared with Rest, the values of noradrenaline were significantly higher at the remaining sampling times during cycling in both the CE and IE (*p* < 0.05 for all).

For dopamine (Figure 3c), there was a significant interaction (*F*_3.78, 22.68_ = 5.91, *p* < 0.01) and significant main effects for [type] (*F*_1.58, 9.47_ = 7.31, *p* < 0.05) and [time] (*F*_2.66, 16.01_ = 9.79, *p* < 0.01). The value of dopamine at E100% in the CE was significantly higher than that in both the CON and IE (*p* < 0.05 for both). Compared with Rest, the values of dopamine were significantly higher at E50%, E100%, and A10min in the CE (*p* < 0.05 for all).

For cortisol (Figure 3d), there was a significant interaction (*F*_3.01, 18.06_ = 5.58, *p* < 0.01) and significant main effects for [type] (*F*_1.57, 9.43_ = 11.27, *p* < 0.01) and [time] (*F*_1.95, 11.74_ = 11.27, *p* < 0.05). The values of cortisol in the CE were significantly higher than those in the CON at all sampling timings except Rest (*p* < 0.05 for all), and the value at E25% in the IE was significantly higher than that in the CON (*p* < 0.01). Compared with Rest, the values of cortisol were significantly higher at E25% in the CE, and significantly lower at A10min in IE and at E50%, E100%, and A10min in the CON (*p* < 0.05 for all).

### 3.4. Comparison of Myotrophic and Neurotrophic Hormones

Figure 4 shows the changes of variables related to myotrophic and neurotrophic hormone responses during the three experimental tests. For GH (Figure 4a), there was no significant interaction (*F*_1.27, 7.67_ = 1.07, *p* = 0.35) and no significant main effects for [type] (*F*_1.31, 7.89_ = 1.32, *p* = 0.29) or [time] (*F*_1.15, 6.92_ = 4.27, *p* = 0.07).

For IGF-1 (Figure 4b), there was a significant interaction (*F*_3.64, 21.86_ = 4.01, *p* < 0.05) and significant main effects for [type] (*F*_1.67, 10.04_ = 5.07, *p* < 0.05) and [time] (*F*_2.13, 12.79_ = 11.67, *p* < 0.01). The value of IGF-1 at E50% in the CE was significantly higher than that in the CON (*p* < 0.05). Compared with Rest, the values of IGF-1 were significantly higher at E50% and E100% in the CE (*p* < 0.05).

For BDNF (Figure 4c), there was a significant interaction (*F*_3.73, 22.42_ = 3.08, *p* < 0.01) and a significant main effect for [time] (*F*_2.33, 13.98_ = 8.94, *p* < 0.01) but no significant main effect for [type] (*F*_1.81, 10.92_ = 2.38, *p =* 0.14). The value of BDNF at E100% in the CE was significantly higher than that in the CON (*p* < 0.05). Compared with Rest, the values of BDNF were significantly higher at E50% and E100% in the CE and at E50% in the IE (*p* < 0.05 for all).

## 4. Discussion

The current study aimed to compare responses regarding the degree of exhaustion and trophic effects between continuous and intermittent cycle ergometry at LT intensity. Two main findings were revealed: (1) intermittent cycling at LT intensity caused a lower degree of exhaustion compared with its continuous counterpart; and (2) intermittent cycling at LT intensity did not induce clear acute trophic responses.

HR and LA have been used to evaluate exercise intensity and physiological effort in many studies [8,29]. Likewise, RPE has been used to evaluate psychological effort during exercise. In practical settings, physical and psychological efforts are thought to be potent factors affecting the continuation and cessation of regular exercise regimens [30]. In the present study, the values of RPE in the IE were lower than those in the CE at all measurement times. Exercise intensity of CE was shown to be 47.6 ± 8.1% HR reserve (HRR) and 2.5 ± 0.6 mmol/L of LA at E25% (five min after the start of cycling), which was consistent with the LT. After that time, these values continued to increase, likely because the LT did not reach a complete physiological steady state [31], and resulted in 65.9 ± 14.0% HRR and 4.1 ± 0.7 mmol/L of LA at E100% (i.e., only after 20 min of cycling). In contrast, exercise intensity of IE was also found to moderately increase during cycling, but the HRR ranged from 22.3 ± 5.6% at E25% to 31.2 ± 5.1% at E100%, and LA ranged from 2.2 ± 0.4 mmol/L at E25% to 2.3 ± 0.7 mmol/L at E100%. As mentioned above (Figure 1), IE was accompanied by a number of brief (i.e., 30-s) rest periods during the protocol, which may have led to the observation of a lower level of cycling effort for IE compared with CE in this study.

In comparisons of exhaustion with additional indices derived from blood analyses, IE showed comparable or lower values compared with CE. Specifically, while adrenaline concentration exhibited a similar increase in both exercise conditions, this was not the case for the increase in noradrenaline concentration (i.e., the increment was lower in the IE than in the CE). Furthermore, dopamine concentration increased in the CE, but not in the IE. Regarding cortisol concentration, in contrast to consistently higher values in the CE compared with the CON, concentration increased only immediately after the onset of exercise (E25%) and decreased thereafter in the IE. Concentrations of catecholamines and cortisol are reported to increase with exercise stimulus at and above LT intensity [32,33,34] to assist the adjustment of disturbed homeostasis [34,35]. Previous studies had shown that increased plasma noradrenaline concentrations during exercise were closely associated with heart rate, which is consistent with the results of this study [34]. These observations in the additional indices regarding exhaustion may also reflect different levels of exercise effort between CE and IE.

Contrary to our original hypothesis, intermittent cycling at LT intensity did not induce clear acute trophic responses. In the present study, we focused on concentrations of GH, IGF-1, and BDNF to examine differences in myotrophic and neurotrophic effects between CE and IE. The changes in GH concentration were not significant in both exercise tests, possibly because of the relatively large inter-subject variability of the changes, especially in CE (Figure 4a). IGF-1 concentration only increased in CE at E50% and E100%, which was consistent with the results of previous reports [13]. Whereas IGF-1 responses have been reported to be rapid and to peak at <10 min of exercise time [36], partly due to an increase in mechanical stimulation [37], the 1-min exercise time, or one min of exercise accumulation, was probably insufficient to increase IGF-1 concentration in the IE in the current study. Similar trends were found for BDNF concentration, despite the slight increase found in IE at E50%. On the basis of these observations, it might be reasonable to conclude that some trophic responses are induced by CE but not by IE. Nonetheless, IE is still considered to be a useful measure to maintain and improve energy expenditure with moderate physical and psychological efforts.

One strength of the present study was the use of multiple measures to objectively demonstrate the degrees of exhaustion and trophic effects during the experimental tests, including blood biomarkers based on peripheral cannulation. Several limitations should be noted for appropriate interpretation of the results. First, the subjects in the current study were men, and all were relatively healthy and active, without any of the adverse conditions listed in the exclusion criteria. Thus, our sample may not be representative of the general population within the same age range. Second, the small sample size of this study may have influenced the ability to demonstrate significant trends, and therefore, it should be taken with caution. Third, our observations were limited to the use of the cycle ergometry, and may have differed if other exercise modes were used, such as jogging and running. Forth, the present study addressed trophic effects mediated by GH, IGF-1, and BDNF, and was not able to address other trophic effects.

Taken together, the current results suggest that intermittent exercise at LT intensity might be a feasible mode of exercise to help people start and maintain regular exercise habits and, consequently, is likely to be a useful tool to prevent and/or moderate metabolic and cardiovascular health problems, especially in older people and/or sedentary populations. Future studies will be required to further explore the physiological benefits of intermittent exercise at LT intensity.

## 5. Conclusions

In summary, the present study demonstrated that intermittent exercise consisting of repeated bouts at LT intensity separated by brief rest induces a lower degree of acute stress or exhaustion than its continuous counterpart. The current findings also revealed potential trophic responses in continuous exercise, but not in intermittent exercise. The present findings could be informative for improving preventive and therapeutic exercise interventions in practical settings.

## Figures and Tables

**Figure 1 ijerph-19-07503-f001:**
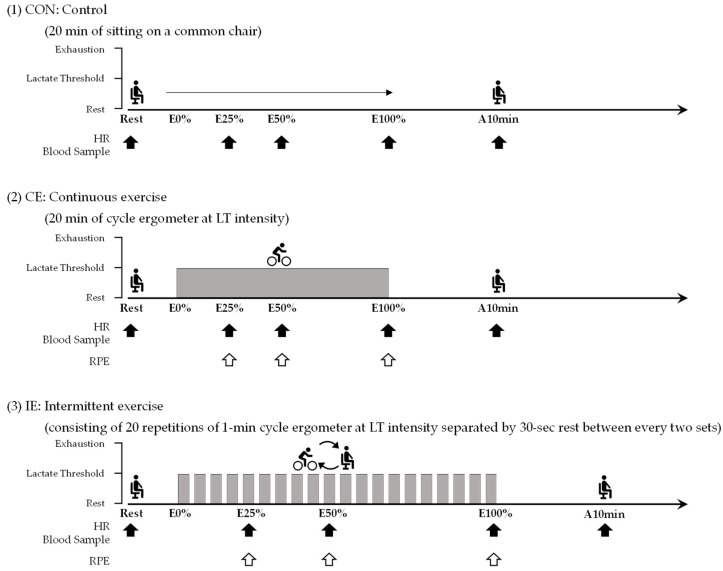
Experimental design: The control (CON) comprised a 20-min period of sitting on a standard chair, continuous exercise (CE) was performed as a 20-min bout of cycling on the cycle ergometer at lactate threshold (LT) intensity, and intermittent exercise (IE) was performed as 20 sets of a one-min bout of cycle ergometer exercise at LT intensity with a 30-s rest between every two sets. HR: heart rate, RPE: Borg rating of perceived exertion. HR measurement and blood sampling were performed 10 min before the onset of cycling (Rest), at the 25%, 50%, and 100% time points of exercise (E25%, E50%, and E100%), and at 10 min after running (A10min), while RPE was at E25%, E50%, and E100%. The closed arrow (
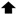
) and open arrow (
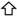
) indicate the timing of HR measurement and blood sampling and the timing of RPE measurement, respectively.

**Figure 2 ijerph-19-07503-f002:**
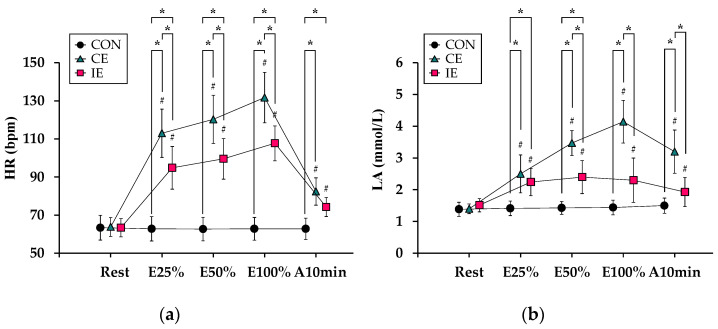
Responses of variables related to a degree of exhaustion (conventional biomarkers): Values of (**a**) heart rate (HR), (**b**) blood lactate concentration (LA), and (**c**) Borg rating of perceived exertion (RPE) are displayed. The circle, triangle, and cube indicate control (CON), continuous exercise (CE), and intermittent exercise (IE) values, respectively. *: *p <* 0.05 indicates [type]. #: *p* < 0.05 indicates [time] with Rest.

**Figure 3 ijerph-19-07503-f003:**
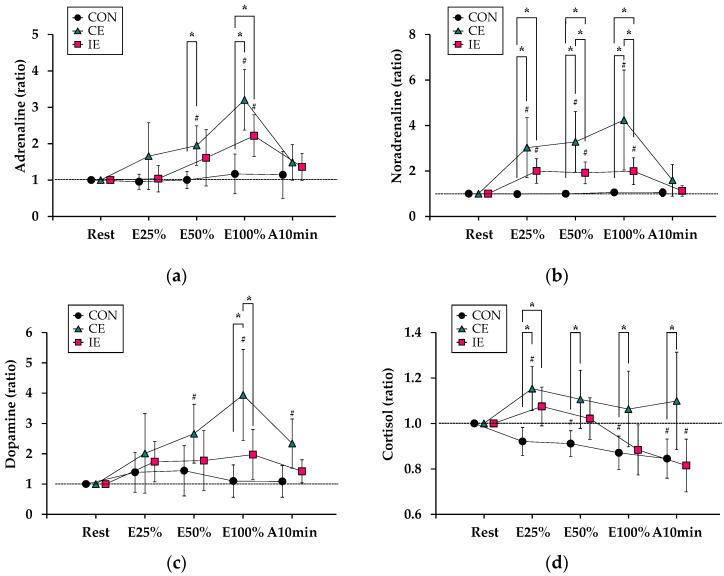
Responses of variables related to additional degree of exhaustion (additional biomarkers): Relative changes of (**a**) adrenaline, (**b**) noradrenaline, (**c**) dopamine, and (**d**) cortisol are shown. The circle, triangle, and cube indicate the control (CON), continuous exercise (CE), and intermittent exercise (IE) values relative to those at Rest, respectively. *: *p* < 0.05 indicates [type]. #: *p* < 0.05 indicates [time] with Rest.

**Figure 4 ijerph-19-07503-f004:**
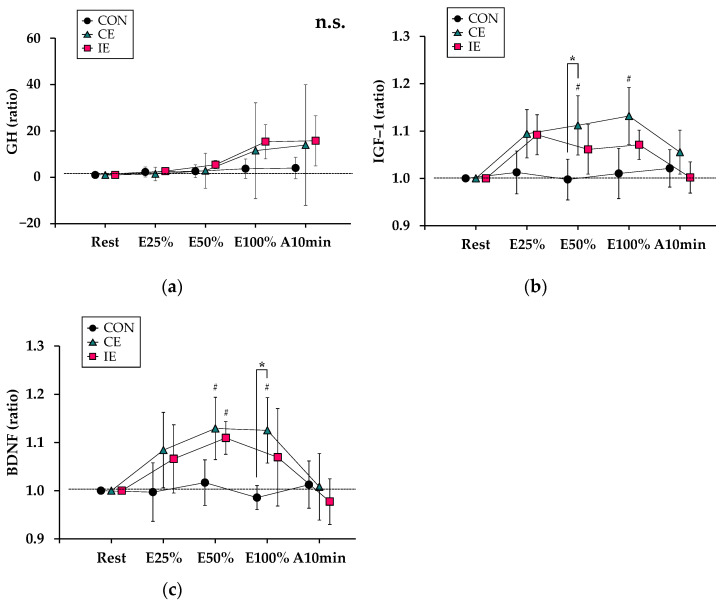
Responses of variables related to myotrophic and neurotrophic hormones: Relative changes of (**a**) growth hormone (GH), (**b**) insulin–like growth factor (IGF) –1, and (**c**) brain–derived neurotrophic factor (BDNF) are shown. The circle, triangle, and cube indicate control (CON), continuous exercise (CE), and intermittent exercise (IE) values relative to those at Rest, respectively. *: *p* < 0.05 indicates [type]. #: *p* < 0.05 indicates [time] with Rest.

**Table 1 ijerph-19-07503-t001:** Characteristics of the study subjects (*n* = 7).

	Means ± SD
Age (years)	59.3 ± 10.4
Height (cm)	168.5 ± 4.2
Weight (kg)	68.9 ± 5.1
Body mass index (kg/m^2^)	24.3 ± 2.3
Cycling load at LT intensity (kp)	1.8 ± 0.4
Cycling METs at LT intensity	6.3 ± 1.0

LT: Lactate threshold, METs: Metabolic equivalents.

## Data Availability

Data are not publicly available as the data contain confidential clinical information of the subjects, although they are possibly made available from the corresponding author upon reasonable request.

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
