# Peer review of "Intermittent Exercise at Lactate Threshold Induces Lower Acute Stress than Its Continuous Counterpart in Middle-to-Older Aged Men"

_ijerph, 2022, doi:10.3390/ijerph19127503_

Round 1
Reviewer 1 Report
Ø The study “aimed to compare responses regarding the degree of exhaustion and trophic effects (l. 72) between continuous and intermittent aerobic exercise using cycle ergometry at LT intensity (l. 73) over the same cycling work volume”.
Ø The authors address in a satisfactory way the gap in this issue in the introduction section and clearly refer to the aim of their study in the last paragraph.
Ø The methodology design (experimental tests, measurements, etc) is clear and is reported in detail.
Ø The results of the study are presented in a satisfactory way
Ø The authors present the main findings of their study in the first paragraph of the discussion section.
Ø The authors should add to the limitations of their study the fact that the sample size was too small. (line 296)
Ø Propose for further research is mentioned correctly on line 308
Reviewer 3 Report
General
First of all, the reviewer would like to thank the authors for their work and efforts in trying to improve sports science knowledge.
The authors aimed to compare responses regarding the degree of exhaustion and trophic effects between continuous and intermittent aerobic exercise using cycle ergometry at LT intensity over the same cycling work volume. The article is well written.
Abstract:
The results of the abstract state: In terms of trophic effects, both IGF-1 and BDNF increased only in CE, and not in IE.
However, in the results section it is stated that: Compared with Rest, the values of BDNF were significantly higher at E50% and E100% in the CE and at E50% in the IE (p < 0.05 for all).
The abstract should be consistent with the results set out in the paper
Introduction:
Well-written and justified
Material and methods:
Legend figure 1. Symbols missing between brackets. (The closed arrow ( ) and open arrow ( )
Results:
Well designed and written
Discussion:
It should be included as a limitation, that the sample is small so the results should be taken with caution.
References:
I recommend that the list of references be updated. One third is more than twenty years old.
